# Indigenous and Non-Indigenous Theories of Wellbeing and Their Suitability for Wellbeing Policy

**DOI:** 10.3390/ijerph191811693

**Published:** 2022-09-16

**Authors:** Tamara Mackean, Madison Shakespeare, Matthew Fisher

**Affiliations:** 1College of Medicine and Public Health, Flinders University of South Australia, Bedford Park, SA 5042, Australia; 2The George Institute for Global Health, Newtown, NSW 2042, Australia; 3Stretton Institute, The University of Adelaide, Adelaide, SA 5005, Australia

**Keywords:** wellbeing theory, typology, public policy, indigenous wellbeing, theory of change, social determinants of mental health

## Abstract

A growing interest among governments in policies to promote wellbeing has the potential to revive a social view of health promotion. However, success may depend on the way governments define wellbeing and conceptualize ways to promote it. We analyze theories of wellbeing to discern twelve types of wellbeing theory and assess the suitability of each type of theory as a basis for effective wellbeing policies. We used Durie’s methodology of working at the interface between knowledge systems and Indigenous dialogic methods of yarning and deep listening. We analyzed selected literature on non-Indigenous theories and Indigenous theories from Australia, New Zealand, Canada and the United States to develop a typology of wellbeing theories. We applied political science perspectives on theories of change in public policy to assess the suitability of each type of theory to inform wellbeing policies. We found that some theory types define wellbeing purely as a property of individuals, whilst others define it in terms of social or environmental conditions. Each approach has weaknesses regarding the theory of change in wellbeing policy. Indigenous relational theories transcend an ‘individual or environment’ dichotomy, providing for pluralistic approaches to health promotion. A broad theoretic approach to wellbeing policy, encompassing individual, social, equity-based and environmental perspectives, is recommended.

## 1. Introduction

Health promotion supports people to move toward a state of physical, mental and social wellbeing [1]. The 1986 *Ottawa Charter for Health Promotion* [1] defined health promotion as healthy public policy across sectors to create healthy social conditions, empower communities and reorient services. This view is consistent with evidence on social determinants of health (SDH) [2], which calls for a whole-of-government approach to health [3]. Indigenous, holistic conceptions of wellbeing inform models of health promotion centred on relatedness to the natural world, strong cultures, family and kinship, and socio-political empowerment [4,5], which arguably would also be of benefit to non-Indigenous populations [6]. Again, this is strongly consistent with evidence on social determinants of Indigenous people’s health and wellbeing, which include control over life circumstances, strong cultural identity, and connection to country [7,8,9]. However, health policies in many societies [10] and in Australia [11] have focused predominantly on remedial medical care, and health promotion is limited to individualised ‘lifestyle’ strategies intended to change behaviours related to diet and exercise, drug and alcohol use, and smoking [12].

Recently, governments in Australia [6,13,14] and elsewhere [15,16] have shown increasing interest in developing policies to promote human wellbeing and measure progress. Some wellbeing frameworks adopted by governments [14,16] recognise known SDH, suggesting that this may be a new opportunity to push public health policy beyond the dominance of medical care and behaviourist health promotion. However, biomedical and individualised behavioural approaches to health maintain a strong hold on policy makers [12]. Furthermore, the concept of wellbeing itself can be theorised and pursued in largely individualised ‘lifestyle’ terms [17].

The concept of wellbeing is also centrally concerned with psychological health [18], and effective policy action on wellbeing has potential as a response to the growing burden of mental ill-health [19]. Common forms of mental illness figure prominently in the burden of disease worldwide [20] and in Australia [21]. Again, if the aim is wellbeing, there is a need to shift policy emphasis away from remedial treatment of individual illness [11] toward strategies to promote psychological wellbeing, including by addressing social determinants of mental health (SDMH) [19].

Recognised SDMH include income, education, employment conditions, housing status, social relatedness, and early childhood conditions [22,23,24]. Chronic psychosocial stress is a key factor mediating the relationship between adverse social conditions and poor mental health [22,25]. In many countries [26], the rates of common forms of illness such as depression and anxiety are higher among groups subject to socioeconomic disadvantage. Conversely, social and environmental factors such as positive social relationships [23] and engagement with nature [27,28] support positive mental health.

In contemplating whether growing governmental interest in wellbeing can shift public policy toward an eco-social view of health and health promotion, a key question arises as to what theoretical stance on wellbeing governments will adopt for the purposes of formulating wellbeing policies. As Taylor [29] observes, if governments want to promote or measure wellbeing, they must surely hold some conception of what wellbeing is and how it can be improved. This presupposes theory in some form. More broadly, theory is regarded as intrinsic to all public policies or programs in the form of an explicit or implicit *theory of change* that defines an issue as a policy problem [30] and makes a ‘predictive assumption about the relationship between desired changes and the actions that may produce those changes’ [31] (p. 1). A theory-of-change approach to policy evaluation makes such assumptions explicit so that they can be tested [31]. Thus, in summary, ‘theory’ for public policy is an implicit or explicit set of ideas intended to define a particular phenomenon as a matter of policy concern and (purportedly) explain how and why taking certain actions will lead to intended outcomes [32,33]. These theories of change *matter* for policy precisely because they serve to define and delimit policy makers’ perceptions, plans and strategies concerning the nature and causes of social problems and the kinds of strategies that ought to be applied to remedy or reduce those problems [6,11,12,30,34].

Therefore, considering growing governmental interests in wellbeing policies, the potential for differing approaches, and evidence on social determinants, this research asks several related questions:What types of wellbeing theory are available to policy makers?What theory of change are differing types of wellbeing theory likely to present to policy makers, for purposes of application in wellbeing policies?How and to what extent are the theories of change implicit in different types of wellbeing theory—if taken up in policy—likely to address social determinants of mental health, and of Indigenous health, including the role of stress?

Furthermore, our previous work [6] examining Indigenous and non-Indigenous theories of wellbeing in Australian health policies argued that, in addition to their importance for Indigenous peoples, Indigenous theories have the potential to inform more holistic approaches to health and wellbeing promotion in the population at large. Therefore, we include both Indigenous and non-Indigenous theory in our analysis of theory types and address an additional question:4.How are Indigenous and non-Indigenous theories of wellbeing similar or different, and what is the potential for integration?

In this paper, we first analyse selected literature on Indigenous and non-Indigenous theories of wellbeing to present a typology of wellbeing theory, addressing question 1 above. We then critically assess the suitability of each type of theory to inform effective policy to promote public wellbeing; based on an appreciation of the role of theory in public policy and recognition of evidence on social determinants of health, of mental health and of Indigenous health; addressing questions 2 and 3. Toward the end of Section 3, we address question 4. In Section 4, we then consider the implications of our findings for future public policy to promote Indigenous and non-Indigenous people’s wellbeing and reduce inequities in wellbeing.

## 2. Materials and Methods

This research was comprised of methods to analyse and interpret theories of wellbeing and assess their suitability for public policy to promote wellbeing. Building on previous work [35], we adopted Durie’s general approach [36] of conducting research at the interface between Indigenous and non-Indigenous knowledge systems to develop shared knowledge. We applied Durie’s principles of mutual respect, shared benefits, human dignity and discovery [36] to undertake the research as a process of collaborative enquiry and mutual capacity-building between Indigenous (TM and MS) and non-Indigenous (MF) authors.

To develop a typology of wellbeing theory, we decided on a method of literature review and undertook initial scoping and discussion between the authors about a suitable style and approach. As we describe below, we identified many whole fields of literature pertaining to human welfare or wellbeing, broadly defined. It became obvious that any form of systematic review of this full range of material was not feasible and, indeed, would not serve our purpose. Instead, we decided to undertake a narrative review of a far more limited body of purposively selected literature, with selection strategies designed to pick materials most suited to identifying *types* of Indigenous and non-Indigenous wellbeing theory. For the purposes of the exercise, we defined wellbeing theory as a set of ideas, presented in written form, that at a minimum offers a definition of wellbeing (i.e., that which constitutes human wellbeing), and may also offer an explanation of the aetiology of wellbeing (i.e., how it comes about) [32,33].

Our selection strategy had three main elements, conducted in parallel. First, to select non-Indigenous theory, we reviewed several recent journal articles explicitly focused on wellbeing theory, where well-known authors in the field summarized theory types [18,37,38,39]. We drew on these to develop an initial, provisional list of theory types and identify and compile key academic journal articles or scholarly reports cited as leading, contemporary (published on or after 1970) exemplars of a particular type of theory. We then drew on several anthologies of wellbeing theory (e.g., [40,41]) to refine and extend our initial list of theory types and compile additional exemplar articles. We then examined reference lists from these compiled exemplar articles, used key word searches in Scopus and Google Scholar and searched websites of relevant journals such as the *International Journal of Wellbeing* to identify and compile any other key works that may have been missed.

Second, to select Indigenous theories, we looked for scholarly written works of two types: (a) works substantially focused on describing Indigenous peoples’ conceptions of wellbeing, in relation to a specific Indigenous cultural group; or (b) works expounding a theory of wellbeing, authored or co-authored by Indigenous scholars, with a particular focus on the wellbeing of Indigenous peoples. To limit the amount of material selected, we limited searches to grey literature or academic articles focused on the wellbeing of First Peoples in Australia, Canada, Aotearoa (New Zealand) and the USA. These countries are also similar in that they are ‘developed’ high-income countries with colonized Indigenous peoples. Authors TM and MS’s understanding of literature on Aboriginal and Torres Strait Islander health and knowledge systems also informed our selection.

Third, based on our interests in and knowledge of public health, health promotion and public policy [6,11,12,35], we selected a limited number of key articles discussing key theorized conceptions of health present in public health literature and practice, in particular, the biomedical and behavioral conceptions of health recognized as shaping public health policy [10,12,42]. We reasoned that any classification of wellbeing theory types relevant to policy would be lacking without these included. We also included selected literature on theories of ‘human development’ [43] based on a connection with capability theory [44] and international development policies.

The total number of items included in the review was 333. Non-Indigenous theoretical considerations of wellbeing were discussed in 128 articles compared to 205 Indigenous specific documents, the majority of which focused primarily on Aboriginal and Torres Strait Islander perspectives. To chart and analyze the material, we grouped articles under types of theory. Author MS skim read all articles and read in depth those identified as formative or widely cited theoretical works under each type. To conduct a more in-depth analysis of theory types and develop the typology, the team developed a framework of six questions: how is wellbeing defined; what elements of wellbeing are described; where wellbeing is seen to be manifested (e.g., within the individual, in the relationship between individual and their social conditions); whether the theory linked to measures of wellbeing; whether the theory recognizes/explains social determinants of mental health/wellbeing; and whether it recognizes/explains the role of stress. Author MS applied this framework to compile analysis summaries that identified key articles from leading theorists under each type, and cited passages of text from these to show how they addressed each of these six questions.

To construct our typology, we then applied our interface methodology in meetings of the research team. We used Indigenous dialogic methods of open-ended yarning and deep listening. Yarning is an Indigenous Australian social practice of conversation to share stories and ideas and develop knowledge. In the context of our research and interface methodology, this involved team members sharing their respective Indigenous (TM and MS) and non-Indigenous (MF) perspectives on the analysis summaries prepared by MS to deepen our analysis on key issues such as: the substance of theories characterizing each type; distinctions between types of theory; similarities and differences between Indigenous theories including how wellbeing has been theorized in post-colonial contexts; and similarities and differences between Indigenous and non-Indigenous theories. In keeping with Durie’s principles [36], in these processes of developing knowledge through yarning, Indigenous and non-Indigenous perspectives were treated as expressions of equally legitimate knowledge systems.

As the analysis proceeded, it became clear that while some individual theorists worked wholly or mostly with one conception of wellbeing (with eudemonic and hedonic conceptions most common in our sample), others (knowingly or not) combined several such concepts. The latter approach reflects a growing recognition in the field that—one way or another—wellbeing is a multi-faceted phenomenon [38]. Thus, we concluded that the types we were looking for were not to be defined as the shared views of a distinct groups of theorists as such, but as instances of theorizing that shared a distinct conception of what wellbeing fundamentally *is* or what it *consists in*. The typology presented below reflects this conclusion.

As the typology resolved into its final form, we continued to use our ‘yarning at the interface’ process of analysis to assess the suitability of different theory types to inform wellbeing policy. In relation to each theory type in our typology, we developed shared perspectives on two key issues relevant to research question 3 noted earlier: the theory of change that each theory type was likely to offer policy makers; and the extent to which each type of theory, as exemplified in the selected literature; formed a basis for action on social determinants of mental health.

We do not report in detail on the various measurement tools we found linked to instances or types of theory, but we do discuss the implications of analyses examining associations between data on wellbeing indicators and measures such as socioeconomic status, gender, Indigeneity or location.

No ethical approvals were required for the research.

## 3. Results

In our initial scoping of literature, it was readily apparent that theories or concepts of human wellbeing or welfare are defining features of many fields of ‘Western’ academic thought and research including but not limited to political and moral philosophy, public policy, economics, law and human rights, sociology, theology, psychology, education, ecology, public health, medicine and development theory [40,41,43]. Similarly, Indigenous theories and concepts of wellbeing are present in literature focused on issues such as health and healing [45], ageing [46], ecology [47] and critical sociology and politics [48] just to name a few. The field of psychology alone features notable theorists of wellbeing such as Maslow, Rogers, Erikson and Allport [49]. Wellbeing or welfare is an established topic of long-standing interest in philosophy [50,51], including in ‘Eastern’ philosophies such as Buddhism or Taoism [52]. Consequently, literature on wellbeing is vast and diverse and impossible to thoroughly explicate in any one review. However, this glimpse of the breadth of the topic is still instructive; indicating that diverse concepts of human welfare or wellbeing are fundamental in shaping our societies. An appreciation of this diversity led us to limit the scope of the review in the manner described above. Consequently, we acknowledge that the typology presented below is also limited.

### 3.1. A Typology of Wellbeing Theory

The types of theory we identified are listed and defined in Table 1. Substantive points of difference between types included whether theorists focus on individuals’ subjective experiences, personal attributes or choices as the locus of wellbeing, or define wellbeing in terms of the isolate individual, the individual-in-environment, or in a more collectivist, ecological or relational sense. These points of difference were also reflected in the various measurement tools we identified, linked to particular theories.

Within any one concept, individual theorists differed in relation to the particular subjective states, personal attributes, behaviors, or objective conditions which were seen to constitute that kind of wellbeing; however, we do not ‘unpack’ all of these nuances here. In formulating the typology in Table 1, we also recognized that holistic Indigenous theories of wellbeing do not necessarily endorse neat analytical distinctions between, say, personal, spiritual, cultural and ecological aspects of wellbeing, but rather may see these as inter-related and inter-dependent elements of larger whole. We will discuss these issues further in what follows.

### 3.2. Non-Indigenous Theories and Their Suitability to Inform Public Policy

We now discuss various types of theory in relation to their suitability to inform effective public policy to promote public wellbeing. Our analysis is based on the premises outlined in the introduction and appreciation of similarities and differences between types on the question of where wellbeing is understood to be manifest.

#### 3.2.1. Individualized Theories

We found that biomedical, behavioral, hedonic, eudemonic, and preference satisfaction theories define wellbeing as a fundamental property of individuals, whether in terms of their physiology, behavior, choices, subjective states or personal attributes and abilities. All theories that we found to align closely with these types were non-Indigenous in origin. However, Indigenous theories may contain elements consistent with one or more of these types. In relation to each of the first four types noted, it was clear from the literature that defining concepts of wellbeing are commonly used to design and apply instruments that (notionally) measure that form of wellbeing, or some aspect of it. Alternatively, existing data may be adapted, where it is seen as a valid proxy indicator of (a theorised aspect of) wellbeing [54]. It appeared that some such measures are specifically intended to fulfil calls for additional measures of social welfare, alongside conventional measures such as gross domestic product (GDP) [55].

Not surprisingly, individualized theories of wellbeing were aligned with measurement tools designed to gather corresponding forms of information *about* individuals, such as markers of biomedical health status, health behaviors, states of mind, or personal attributes. For example, hedonic theories of subjective wellbeing are frequently linked to instruments that ask people questions about their subjective states of happiness or satisfaction [56]. Furthermore, data gathered using such instruments are sometimes aggregated and combined with other population data to identify statistical correlations between social factors such as income or education and the particular wellbeing measures in question [54,56].

Just as with other forms of public health research, statistically significant correlations between these variables could be used as prima facie evidence that this or that socioeconomic factor may be acting as a determinant of wellbeing (as defined in the corresponding theory). However, according to our approach, a key question (apropos the *value of theory for policy*) is what kind of explanation, if any, the ‘originating’ theory can provide about such correlations, and thereby how it will define the problem and inform a theory of change for policy [32]. As we argue below, the individualized nature of these five kinds of theories is crucial in answering this question.

With a biomedical view of health, clearly biomedical measures of morbidity or mortality are used with other data to identify socially patterned differences in those measures [2]. Furthermore, a biomedical view can be applied to explain the biomedical aspects of such differences, e.g., the effects on tobacco smoke on lung tissue as an explanation of higher rates of lung cancer among smokers. However, we would argue that the theory of change implicitly presented is to change the trajectory of disease, or the symptoms or vectors of increased disease risk, by applying biomedical interventions. This is an individualized approach, simply because the perceived problem to be solved lies within the individual body, in the form of ‘disease’. Thus, the preferred approach to evidence on health inequalities would be to increase access to such interventions for those groups at higher risk or with established disease, and this approach is indeed widely evident in health policy [10,11]. A biomedical conception of wellbeing, in and of itself, does not offer any theory of change for actions on SDH to promote wellbeing, nor does it offer a basis for strength-based and ‘whole person’ approaches to Indigenous mental health [57]. It may be used to recognize internal physiological effects of chronic stress, but again, as theory of change, it will limit its recommendations to biomedical interventions to ‘treat’ these effects directly.

Similarly, behavioral concepts of wellbeing may be statistically linked with social factors. However, again, the conception of wellbeing necessarily locates presumptive deficiencies in wellbeing, or increased risks of disease, wholly within the individual and their behavioral choices. Thus, the theory is likely to present a theory of change to policy makers whereby individuals (perhaps those in high-risk groups) are ‘assisted’ in some way to acquire the health literacy required to choose rationally to modify relevant behaviors [12]. Again, there is nothing here to problematize or stimulate action on SDH or SDMH as such.

With hedonic theories, such as subjective wellbeing (SWB) theory, again, data from SWB measures can be combined with other data to describe socially patterned differences in SWB, between groups or over time [56,58]. However, the theory itself, we surmise, does not contain any explanation for why such differences occur, which might strengthen a theory of change for policy to improve SWB. By definition, SWB states are just ‘there’ to some degree, or not, according to an individual’s introspection. It seems to us that any theory of change here for policy makers to improve SWB will have to borrow ideas from other wellbeing conceptions such as improving eudemonic attributes such as resilience, or improving living conditions [58].

The story for eudemonic theories is similar: they do inform wellbeing measures which are sometimes used to examine social patterned differences in theorized wellbeing attributes [54]. However, once again, the individualized nature of the underlying theory is likely to present a theory of change based on improving individuals rather than modifying relevant environmental conditions.

We did find evidence in the literature that measures of hedonic or eudemonic wellbeing have been correlated with markers of stress arousal such as salivary cortisol levels [49], thus acknowledging the role of stress as causal factor in the onset of mental ill-health conditions. However, in our reading, this interest in the role of stress has not resolved into an explanatory theory about *how* stress works as a mediator of social–environmental impacts on stress, including how and why (under what kinds of conditions) the role of acute stress in everyday social cognition ‘converts’ to chronic stress [19]. Thus, there is no basis to explain how certain socio-environmental conditions assist wellbeing by supporting the self-regulation of the stress ‘load’ [19].

Preference satisfaction theories of wellbeing are a form of hedonic theory. However, we identify them as a distinct type because they have a particular place within contemporary free-market economic theory. To wit, they justify a view that markets serve wellbeing because, within the processes of supply and demand, individual acts of consumption are presumed to be satisfying preferences [18]. We did not identify any literature measuring preference satisfaction as such, but arguably measures such as income or per capita GDP are regarded as proxy measures [48]. The theory of change on offer to policy makers, obviously enough, is that measures to stimulate consumption—such as reduced taxes—will increase wellbeing. In our assessment, preference satisfaction theory is just simplistic and explains almost nothing about the complex nature or aetiology of wellbeing as revealed by empirical research [55]. Even on its own limited terms, we know that, above a certain level of income per capita, gains in national measures of SWB tail off rather than continuing to match growth in incomes, as preference satisfaction theory would predict [56]. In a country such as Australia, preference satisfaction explains precisely nothing about why, despite historically high levels of material consumption, one in every five people is subject to a mental health disorder in any 12-month period [59].

Therefore, we conclude that these forms of individualized theory do not offer a sound theoretical basis to inform policy on how to promote wellbeing, taking account of social determinants of mental health and the mediating role of stress. Why? because while they may acknowledge stress physiology as a potential risk factor for mental health, they do not offer clear theoretical explanation of the interactions and mechanisms operating between the individual and their social conditions which cause or prevent chronic stress. In one way or another, the make-up of these types of theory lacks the conceptual equipment to formulate such an explanation.

#### 3.2.2. Living Conditions Theories

This category includes a number of different theories, which share a view of wellbeing as constituted by individuals having *access*, or being *exposed*, to certain kinds of favourable social or environmental conditions, and ‘using’ or gaining from these (in some sense) to their own benefit [43]. In other words, living conditions theory (as we have named it) is marked by positing a set of conditions regarded as needful for human health/wellbeing, development or happiness. Such conditions may be conceptualized as essential needs. They may be implied in the definition of conditions identified as bad for health or wellbeing. Theory may be formulated based on generalizations arising from empirical research examining, for example, correlations between markers of social or environmental conditions and health outcomes. Equally, theory may inform the design and application of tools to measure access to the relevant conditions. Again, it was mainly non-Indigenous theories identified as fitting the type, but Indigenous theories may certainly incorporate consideration of certain objective conditions.

Defining living conditions theories as a type of wellbeing theory is liable to an objection that they are about favourable *conditions* and not wellbeing itself. We would say, rather, that here, wellbeing is defined within a construct of individual-in-environment. Different theories variously use one or more of the individualized conceptions noted above to fill in the personal ‘side’ of that construct. For example, an SDH view tends to theorize wellbeing in terms of benefits for physical or mental health status derived from access to favourable conditions such as adequate income, education, housing, health or secure well-paid employment [2,19]. The health status in question tend tends to be conceptualized in biomedical terms (morbidity or mortality), reflecting the underlying role of epidemiological evidence in the formulation of SDH theory [19]. However, concepts in the literature such as ‘sense of control’ draw from more of a eudemonic flavour [60]. Indexes of wellbeing (or of ‘health’, ‘human development’ and so on) may just measure a prescribed set of social or environmental conditions or combine these with other eudemonic or hedonic measures [61].

Living conditions theories offer a theory of change to policy makers that—naturally enough—is focused on improving the availability of a defined condition or conditions, or improving, as it were, the quality of a salient condition to which people already have access, on the premise that this will protect or improve population wellbeing or reduce inequities in wellbeing. There is the potential there to strengthen theory by drawing in evidence from other disciplines to *explain* mechanisms and pathways that mediate the causal relationship between conditions and outcomes. Such explanations could improve policy makers’ conceptualization of the problem and appropriate actions to address it, that is, their theory of change [34], but are often lacking. SDH literature commonly invokes stress as a likely mediator of societal impacts on physical or mental health [2], but a clear theorized explanation of how, why and in what circumstances stress plays this role is, again, often limited or missing [19].

#### 3.2.3. Capability Theory

Capability theories are also interested in socioeconomic or environmental conditions favourable to wellbeing but approach matters in a somewhat different way, emphasizing not just access to certain conditions but the kinds of activities a person’s conditions or gender make possible [62]. According to our review, non-Indigenous theorists such as Amartya Sen and Martha Nussbaum have led the initial formulation of capability theory. However, Indigenous theorists are using and adapting capability theory to develop theory and associated measures of Indigenous peoples’ wellbeing and use these as a basis for community action or policy advocacy [63,64].

As a basic outline, capability theory is about a person having both personal resources and enabling social conditions, which enables them to choose from a variety of options for living, and subsequently to live a life that—to use Amartya Sen’s cryptic phrase—they ‘have reason to value’ [65]. The concept of ‘capability’ resides in this combination of conditions, freedoms and actions. Sen distinguishes between ‘functionings’ defined as ‘the various things that (a person) manages to (actually) do or be in leading a life’ and ‘capabilities’ as ‘the alternative combinations of functionings the person can achieve’ [44]. He has declined to specify any universal capabilities, arguing that specification should be resolved within particular contexts. The question of where an idea of wellbeing resides in Sen’s theory is difficult to discern, whether in capabilities or functionings, or in the wider social structures ‘surrounding’ these things [66].

Another capability theorist, Martha Nussbaum, has defined a list of basic capabilities [67]. The list can also be regarded as a form of eudemonic theory, an effort to specify essential elements of a life well lived. However, the concept of ‘ability’ again implies a combination of favourable conditions and personal attributes or actions.

In terms of policy, our assessment is that Sen’s framework is formulated in complex, abstract terms (as philosophy) and would not offer *any* theory of change to policy makers until ‘filled in’ with some definite conception of valued capabilities or functionings, which can then become the object of policy action or evaluation. However, arguably, the value of his open-ended approach is to enable this to be performed in ways specific to particular contexts. Yap and Yu [64] offer an example of such ‘operationalizing’ of Sen’s framework, among and with the Yawuru people of northern Western Australia. They say the value of the theory for them lays in the recognition of individual and collective *agency* as an essential element of Yawuru wellbeing, consistent with a right to self-determination.

The applicability of Nussbaum’s theory seems to lie with thinking about essential capabilities as fundamental entitlements or rights [67]. More generally, for the purposes of policy for wellbeing, we wonder whether the appeal of capability theory might bear a more straightforward interpretation: simply that the social conditions required to support wellbeing will include opportunities for agency and choice. Recognition of agency as an important element of wellbeing is not peculiar to capability theory [68]. There is nothing in Sen’s original formulations of capability theory—in the mode of philosophy or economics—that offers any basis to explain social determinants of mental health.

#### 3.2.4. Indigenous Theories

We found that Indigenous theories are preeminent in theorizing the communitarian, cultural and ecological concepts of wellbeing identified in our typology. Indigenous theories of wellbeing, and associated measures, are being elucidated using contemporary methods of scholarly research and exposition but are also grounded in traditional knowledges derived from Ancestral experiences and continuity and connection between land and culture, spirit, body and mind [69,70].

Whilst conceptions, experiences and practices of wellbeing differ between Indigenous cultures [71]—within or between countries—there are significant shared perspectives [36]. Wellbeing is commonly understood in holistic terms as encompassing multiple aspects of life including individuals, family and kinship, community, ancestors and future generations, country or land [70,71], and the physical world [69,70,71,72,73,74,75]. Thus, wellbeing is not defined as purely a state of individual being, nor as merely an outcome of favourable conditions, but as a manifestation of *interrelationships* between individual, collective and environmental domains sustained through time. Knowledge and wellbeing are seen to arise through experiences of country or land as a living entity [69,71,75]. Individuals are seen to draw nurturance from and carry responsibilities to their social and ecological worlds. A deep and enduring connectedness with the natural world—lands, waters, other living things—is commonly understood as fundamental [36,76]. This ecological view of the interrelatedness of human existence, knowledge, culture, country or land and wellbeing [71,75] is sometimes described as ‘relational wellbeing’ [72,73] or ‘relationality’ [74].

Indigenous cultures are also seen as essential to wellbeing, encompassing forms of philosophy and ethics, empirical knowledge, teaching/learning [75], ceremony and other practices grounded in holistic and relational worldviews. In a similar vein to our study, Salmon et al. [77] reviewed literature from Australia, USA, Canada and Aotearoa and identified five dimensions of Indigenous cultures commonly recognized as essential for wellbeing: connection to country; Indigenous beliefs and knowledge; Indigenous language; family, kinship and community; cultural expression and continuity; and self-determination and leadership. It is apparent how these dimensions encompass the communitarian, cultural and ecological conceptions of wellbeing identified in our typology.

Indigenous conceptions, experiences and practices of spirituality are also commonly embedded within a holistic and relational worldview, and consistent with our proposed definition of spiritual wellbeing. For example, as Poroch et al. [78] describe, Australian Aboriginal spirituality ‘connects past, present and future… [and] recognizes people’s relationships with each other, the living… and non-living… life forces, premised by an understanding or experience of their place of origination.’

In colonized societies, Indigenous theorists have developed new critical theory on the welfare of Indigenous peoples, incorporating concepts such as decolonization, cultural safety, self-determination and Indigenous rights [35,79]. We will not discuss these in detail but suggest that each in its own way is grounded in the holistic, relational, cultural conceptions of wellbeing described above [35,70,75]. These critical perspectives also highlight ways in which white, Western conceptions of human nature and welfare embedded in political systems have shaped and justified violent colonialist processes of genocide, dispossession, subjugation and enforced cultural assimilation. As part of this dynamic, narrow biomedical or behavioral approaches to health can be experienced as a form of colonization when they marginalize Indigenous conceptions and practices of wellbeing [80] and represent Indigenous peoples in deficit-based ways [6].

Indigenous views of wellbeing would appear to be broadly open to the recognition of social determinants of health (SDH), insofar as they already conceptualize individual health and wellbeing as interrelated with social and environmental conditions [70,81]. There can be little doubt that structural social and economic disadvantages to which Indigenous people are subject in many countries also adversely affect their health and wellbeing [82]. However, the fact that SDH thinking continues to rely on biomedical or behavioral measures of health can be problematic for the reasons mentioned previously. Indigenous and non-Indigenous scholars and policy actors seek to address such concerns through development of perspectives and evidence on social determinants of *Indigenous* health, highlighting relational, cultural and rights-based perspectives [7,83,84]. Indigenous theories of wellbeing in colonialist contexts are frequently used to articulate an explicit theory of change involving political support for processes of self-determination that *enact* multiple elements of wellbeing, and broader social and structural reforms to dismantle the racist legacies of colonization [35,85].

#### 3.2.5. Interface between Indigenous and Non-Indigenous Theories

One of the aims of our research has been to examine how Indigenous and non-Indigenous theories of wellbeing are similar or different and consider the potential for theoretic integration. Our wide-ranging review indicates several points where there is the potential for a more integrative approach, combining insight and perspectives from Indigenous and non-Indigenous theory (and associated evidence), which are shown in Table 2 below. For example, in relation to the spiritual concept of wellbeing identified in our typology, we found there are interesting intersections between Indigenous conceptions of contemplative, present-focused states of being, such as the Australian Ngangikurungkurr people’s concept of Dadirri [53], and non-Indigenous concepts of mindfulness or detachment [86,87]. However, we would observe again that, in the example of Dadirri, the orientation is strongly relational and ecological [53], while a concept such as mindfulness can be interpreted in a highly individualized way [88] (although the latter tendency may offend the complexity of the concept in Buddhist theory).

Here, we will not attempt to further tease out the implications of these points of intersection, but simply present them as opportunities for further research and dialogue, working at the interface between knowledge systems [36].

## 4. Discussion

Ideas and beliefs about wellbeing are defining features of human thought in many domains and may have significant real-world effects by shaping human experience and relationships. Our analysis affirms our premise that theoretic perspectives on wellbeing adopted in public policy are likely to really matter because they will shape the definition of wellbeing and theory of change applied and thereby delimit what is actually done or not done. Nowhere is this more evident than in the dominant (and often taken-for-granted) role of biomedical and behavioural theories of wellbeing in shaping contemporary health policies [6,10,11] and marginalizing other social and ecological perspectives.

Recent, renewed policy interest in wellbeing is welcome as an opportunity to revive and extend political and public attention on health promotion, and perhaps to position wellbeing as a core value of governments’ policies in all areas [15]. It comes at a time in which the limitations of biomedical strategies to address a global epidemic of mental ill-being are increasingly obvious [20]. However, the theoretic perspectives used to operationalize policy interests in wellbeing will influence the extent to which these opportunities are realized in practice.

Theories that define wellbeing in terms of the subjective states, attributes or choices of individuals may draw attention to relevant personal aspects of wellbeing, but as theory of change for policy are liable to delimit focus on improving personal resources for coping or ‘living well’ [37,87] rather than improving social conditions. In a neoliberal political environment, such an approach may attract political support despite its limitations [12]. Conversely, living conditions theories certainly offer a basis to act on social conditions but may sideline the role of individual resources and agency. Desired outcomes for wellbeing may continue to be defined narrowly, according to the biomedical or behavioural measures favoured in epidemiology.

Capability theory attempts to combine individual and social perspectives in abstract concepts of capabilities and functionings but may offer little to policy makers until these are specified in particular contexts. The holism and relationality common to Indigenous understandings of wellbeing goes further, transcending the apparent choice between individualist or environmental approaches. Relationality challenges both these approaches to *expand* their explanatory constructs in order to not perpetuate unnecessary and sometimes politicised dichotomies. In historical and philosophical terms, questions are also raised about Western epistemology that has insistently fragmented knowledge into discipline areas [94], which subsequently have been the progenitors of many of the different kinds of wellbeing theory we have examined, along with various tools for purportedly measuring wellbeing. Our analysis suggests that the limitations of many of the different kinds of theory examined, as a basis for policy action on wellbeing, may be tied to this disciplinary fragmentation [95].

Consciously or not, governments are already committed conceptually and fiscally to some of these narrow explanatory constructs such as biomedical, behavioural and preference satisfaction views of wellbeing, built into policy as theory of change or as indicators of success. These commitments are manifestly not working well for human wellbeing and are barriers to flexible thinking and knowledge development. Thus, one would have to question whether governments interested in wellbeing should just switch their allegiance to another similarly narrow construct such as a hedonic or eudemonic theory of wellbeing [17].

Rather, our analysis would suggest the adoption of a more flexible, holistic and open approach to policy for wellbeing, one that is willing to encompass individual, social, equity-based and environmental perspectives. Working at the interface between Indigenous and non-Indigenous knowledge systems [36] is one way to cultivate this kind of approach [6] and enables different perspectives to coexist without inferences of superiority or inferiority. The points of intersection between Indigenous and non-Indigenous theories of wellbeing noted above offer ample material for interface thinking and theoretic synthesis. For example, commonalities between Indigenous and non-Indigenous theory on ecological wellbeing illustrate that, in ways simultaneously material, psychological, cultural and spiritual, human wellbeing is deeply related to other living things and the natural world [53]. This recognition is now essential to securing a viable future.

Again, individualized theories positing personal attributes of wellbeing [37,54] and living conditions theories focused on favourable environments may both have important insights to offer, within such a broader approach to policy. Our analysis does not assert that these various theoretical approaches are wrong as such, only that on their own, as explanatory constructs for policy and social change, they are *too limited*. Healthcare informed by the insight of biomedicine is valuable, of course, but is poorly positioned theoretically to advance the positive dimensions of wellbeing.

However, there is a particular limitation operating across multiple kinds of theory which must be actively corrected. This is the lack of explanation about social determinants of mental health and the role of stress as a mediator of social–environmental impacts on mental health. Policy on wellbeing is less likely to be successful if it proceeds without a clear understanding of *how* the social, economic and environmental conditions we are creating together now are undermining mental health [22] and failing to cultivate wellbeing [19].

Indigenous conceptions of wellbeing also present challenges to some other contemporary norms of politics and public policy. Their communitarian perspectives suggest that communities of all descriptions have a crucial role to play in cultivating the social–relational dimensions of wellbeing, and this challenges norms of designing and funding social service ‘interventions’ from the comfort of ‘head office’. It draws attention to the potential importance of a more localized, ‘place-based’ approach to policy for wellbeing, where local actors are able to exercise more control over the way policy resources are deployed [96,97,98].

Equally, Indigenous perspectives on cultural wellbeing ask policy makers to consider how leadership, shared social norms and public discourse may contribute to or detract from a sense of shared identity and meaning. Non-Indigenous, eudemonic theories commonly identify a sense of meaning or purpose as important for wellbeing but decline to conclude that shared social values and norms might have any role in providing, as it were, meaningful social conditions. Eckersley highlights the role of culture in wellbeing and argues that materialist and individualist values of Western liberal capitalism adversely affect wellbeing by undermining essential ‘psychosocial’ conditions such as a sense of control and supportive social relations [93].

## 5. Conclusions

Renewed policy interest in wellbeing is most welcome. A broad theoretic approach to understanding wellbeing encompassing individual, social, equity-based and environmental perspectives is recommended. Theories of change enacted in public policy should recognize multiple pathways for advancing wellbeing. These can reasonably include strategies to cultivate personal resources for wellbeing, create supportive social environments for mental health and wellbeing, engage and empower communities, and connect with and care for the country. Policy makers should recognize the value of Indigenous perspectives for broader public wellbeing, and facilitate dialogue on wellbeing, working at the interface between knowledge systems.

## Figures and Tables

**Table 1 ijerph-19-11693-t001:** A typology of wellbeing theory.

Defining Concept of Wellbeing	Definition of Wellbeing
**Biomedical**	The absence of defined bodily states of disease or injury, or recovery from such states through the aid of medical treatment; having a normal profile of physiological functioning
**Behavioural**	Informed individuals take rational action to maintain their own health and avoid health risks by adopting healthy behaviours and avoiding unhealthy behaviors; practicing a healthy ‘lifestyle’
**Hedonic**	A state where positive affect (good feeling, pleasure) outweighs negative affect (bad feeling, pain). A subjective feeling or appraisal of being happy or satisfied
**Eudemonic**	Having and exercising personal attributes or behaviours that add up to a well-developed, thriving or flourishing life: e.g., positive attitudes, a sense of purpose, competence, or capacity to cope with life challenges
**Preference satisfaction**	The satisfaction of a person’s (rational, informed) preferences or desires; maximising utility (satisfaction) from goods or services
**Spiritual**	A state of awareness characterised by attention on the present moment, mindfulness, detachment, dadirri (deep listening and quiet, still awareness) [53], a deep sense of connectedness with other people or the natural world
**Living conditions**	Having access to social, economic or environmental conditions that: fulfil basic needs; enable health/wellbeing; or support human development
**Capabilities**	Having conditions and resources that enable one to exercise certain abilities; i.e., to *do things* that together constitute basic dimensions of a good (well-realised) life
**Communitarian**	(Individual participation in) a thriving family, community or society; kinship relations; reciprocity
**Cultural**	Cultural knowledge, norms or values support/enable well-functioning individuals, families, communities
**Ecological**	(Human inter-relatedness with/dependence on) the health of ecological systems
**Holistic**	Individual and community life integrating personal, spiritual, social, cultural, ecological and time dimensions; relational conceptions

**Table 2 ijerph-19-11693-t002:** Points of intersection between Indigenous and non-Indigenous theory.

Indigenous Theory	Non-Indigenous Theory
Dadirri (and similar concepts); deep listening; quiet, still awareness [53]	Buddhist view of mindfulness; contemplation; present-time awareness [86]
Ecological wellbeing; deep relatedness with the natural world	Contact with nature as a determinant of mental health [89]; planetary health [90]
Communitarian wellbeing; relatedness with family and community [75]	Recognition of positive social relationships as an aspect of wellbeing [54]; social relatedness as a determinant of health [23]
Healing conceived in terms of transformative or restorative processes [91] or reconnection to country, land or culture [45,71]	Perspectives on the impacts of and recovery from chronic stress [19]
Culture provides knowledge, values and practices for wellbeing	Sense of meaning and purpose as an attribute of wellbeing [92]; culture as a determinant of health [93]
Holistic, relational understanding of wellbeing	Planetary health [90]; holistic views of health promotion [1]

## Data Availability

Not applicable.

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
