# Peer review of "Indigenous and Non-Indigenous Theories of Wellbeing and Their Suitability for Wellbeing Policy"

_ijerph, 2022, doi:10.3390/ijerph191811693_

Round 1

Reviewer 1 Report

The article is timely and needed. The presentation of the argument between lines 65 and 73 is elegant and clear.

The title and the outlet (namely International Journal of …) suggest a wider international audience, so I suggest adding in the title “in Australia, New Zealand, Canada, and the United States.” There is a very rich literature on South American, Asian, and African, among others, indigenous understandings of wellbeing that differ from those presented. For example, for Asian Indigenous peoples that ideas of harmony differ. Hence, better situate it among the English-speaking countries with similar colonial history from the same nation (Great Britain), where there was legislation that clearly privileged a non-Indigenous policy. 

My general comment relates to the missing of North American Indigenous philosophers such as Gregory Cajate and Brian Yazzi Burkhart, who have written extensively about wellbeing (line 227; lines 425,426; lines 437,438 about relationality). There is also very little citing of work describing wellbeing in Indigenous North America people.

In line 122 and around it, you described your method as dialogical, and narrative. It would be beneficial if you could give an example where being at the interface of Indigenous and non-Indigenous knowledge systems was difficult or presented a challenge. 

Lines 140-141, (a) works substantially focussed …: Ward, Hill et al. (2021) “The land nurtures our spirit”… describe wellbeing for the Innu of Labrador, a cultural group that entered the Indian Act in Canada as late as 2002 (perhaps the last one in Canada). This work can be cited as a North American example of a specific cultural group. At some point, it should be noted that for North American Indigenous peoples, the term used for “country” is “land”, perhaps in a note, to make it accessible to broader audiences outside of Australia.

Line 182, dialogical and open-ended yarning. Please explain what “yarning” for a non-Australian or New Zealand reader is.

Line 207. Health and healing. A North American scholar that worked a lot of Indigenous Healing is Waldram (2013) (transformative and restorative processes…); Ward, Hill et al (2021) Innu process of healing…

Lines 212 to 216 are excellent points and you arrive to them well.

Lines 294 and 303. Is “pattered” a typo? Or did you intend to say “paterrned”?

Line 324” “In our assessment, preference satisfaction theory ….” Can you add a reference that agrees with your assessment? This is quite a string statement.

Lines 397 and 398: “The question of where an idea …” Can you support this statement with a reference? Otherwise, state that it is your opinion.

Lines 430 and 431; 448 to 453: “Country and the physical world” and “conceptions, experiences and practices” The work of Ward et al (2021) with the Innu (the land nurtures our spirit) describes how land is alive and how experiences of land develop their spirituality, arriving at the realization that wellbeing is being connected to all the living world. You might need to explain what country (or land) means. Because it is not just a physical place but the placement of their wellbeing, the placement of the knowledge system, etc. 

Section 3.2.5: The interface

You talk about ‘Western’ and give an ‘Eastern’ example with the Buddhist notion of mindfulness.  Indigenous Cherokee writer Randy S. Woodley has written about the concept of harmony with the land and how it is the same as the Judeo-Christian concept of shalom. Perhaps this or another, are  a better example as something ‘Western’.

Line 529 about the fragmentation of knowledge: Cajete and many others have written about this, add references.

Overall, it is a good and needed work; I would explain the methods better by giving examples of the dialogical process.  There is not enough examples of work outside Australia and New Zealand.

Author Response

Reviewer comment: The article is timely and needed. The presentation of the argument between lines 65 and 73 is elegant and clear.

Author response: Thank you.

Reviewer comment: The title and the outlet (namely International Journal of …) suggest a wider international audience, so I suggest adding in the title “in Australia, New Zealand, Canada, and the United States.” There is a very rich literature on South American, Asian, and African, among others, indigenous understandings of wellbeing that differ from those presented. For example, for Asian Indigenous peoples that ideas of harmony differ. Hence, better situate it among the English-speaking countries with similar colonial history from the same nation (Great Britain), where there was legislation that clearly privileged a non-Indigenous policy. 

Author response: We felt the suggested addition made the title long and unwieldy, but the reviewer’s point is well made so, to clarify the scope of the work ‘up front’, we have changed the abstract to state that Indigenous theories were drawn from Australia, New Zealand, Canada, and the United States.

Reviewer comment: My general comment relates to the missing of North American Indigenous philosophers such as Gregory Cajate and Brian Yazzi Burkhart, who have written extensively about wellbeing (line 227; lines 425,426; lines 437,438 about relationality). There is also very little citing of work describing wellbeing in Indigenous North America people.

Author response: Thank you for the useful suggestion to include more references to work describing wellbeing in Indigenous North American and Canadian peoples, and suggestions to consider the work of Cajete, Burkhart and (as per comments below), Ward et al. and Waldrum. We identified and read realvant articles by Cajete and Burkhart, and the specific articles suggested, by Ward et al. and Waldrum. These are all now cited in the revised text in sections 3.2.4 and 3.2.5 (pp. 10-11 of revised manuscript). Please note other North American and Canadian works already cited from Bockstael et al., Hovey et al., and Hodge et al.,

Reviewer comment: In line 122 and around it, you described your method as dialogical, and narrative. It would be beneficial if you could give an example where being at the interface of Indigenous and non-Indigenous knowledge systems was difficult or presented a challenge. 

Author response: We have revised several main parts of our methods section (on pp. 4-5) of the revised manuscript) to provide more information on the way we applied yarning as a method, and how we used yarning methods to firstly analyse literature on different type of wellbeing theory and develop our typology; and then to assess the suitability of each theory type to inform wellbeing policy. As it happened, the yarning process we applied did not result in any significant challenges arising from conflicting views.    

Reviewer comment: Lines 140-141, (a) works substantially focussed …: Ward, Hill et al. (2021) “The land nurtures our spirit”… describe wellbeing for the Innu of Labrador, a cultural group that entered the Indian Act in Canada as late as 2002 (perhaps the last one in Canada). This work can be cited as a North American example of a specific cultural group. At some point, it should be noted that for North American Indigenous peoples, the term used for “country” is “land”, perhaps in a note, to make it accessible to broader audiences outside of Australia.

Author response: Ward et al. is now first cited on p. 10 of the revised manuscript, where we make the point that ‘conceptions, experiences and practices of wellbeing differ between Indigenous cultures [71].’ In the next sentence we address the reviewer’s comment about the terminology of ‘country’ and ‘land’, and again cite Ward et al. and also Burkhart. In lines 450-452 we say that ‘Wellbeing is commonly understood in holistic terms as encompassing multiple aspects of life including individuals, family and kinship, community, Ancestors and future generations, Country or Land [70, 71], and the physical world [69-75].’

Reviewer comment: Line 182, dialogical and open-ended yarning. Please explain what “yarning” for a non-Australian or New Zealand reader is.

Author response: Text changes in the methods section in lines 176-185 now include an explanation of yarning as applied in our research. 

Reviewer comment: Line 207. Health and healing. A North American scholar that worked a lot of Indigenous Healing is Waldram (2013) (transformative and restorative processes…); Ward, Hill et al (2021) Innu process of healing…

Author response: Thanks again for these suggestions. Both Waldrum and Ward et al. are now cited in Table 2 (pp. 11-12), where we note some Indigenous conceptualizations of healing.

Reviewer comment: Lines 212 to 216 are excellent points and you arrive to them well.

Author response: Thank you

Reviewer comment: Lines 294 and 303. Is “pattered” a typo? Or did you intend to say “paterrned”?

Author response: Thanks for pointing this out. Words have been amended to ‘patterned’.

Reviewer comment: Line 324” “In our assessment, preference satisfaction theory ….” Can you add a reference that agrees with your assessment? This is quite a strong statement.

Author response: Thanks for this suggestion. To support our statement, now on lines 344-346, we have cited the 2009 “Report by the Commission on the Measurement of Economic Performance and Social Progress” by Stiglitz et al. Their chapter 2 on ‘Quality of Life’ presents a critique of a preference satisfaction approach to wellbeing, claiming that it overlooks ‘many of the determinants of human well-being [that] are not monetary resources but aspects of people’s life circumstances.’

Reviewer comment: Lines 397 and 398: “The question of where an idea …” Can you support this statement with a reference? Otherwise, state that it is your opinion.

Author response: We have added somewhat to the sentence, now on lines 417-419, to further clarify our meaning and citee a supporting reference. It now reads as: “The question of where an idea of wellbeing resides in Sen’s theory is difficult to discern; whether in capabilities or functionings, or in the wider social structures ‘surrounding’ these things” [66]. The reference in question is, Jackson WA. Capabilities, culture and social structure. Review of Social Economy. 2005;63(1):101-24.

Reviewer comment: Lines 430 and 431; 448 to 453: “Country and the physical world” and “conceptions, experiences and practices” The work of Ward et al (2021) with the Innu (the land nurtures our spirit) describes how land is alive and how experiences of land develop their spirituality, arriving at the realization that wellbeing is being connected to all the living world. You might need to explain what country (or land) means. Because it is not just a physical place but the placement of their wellbeing, the placement of the knowledge system, etc. 

Author response: We have amended text on lines 455-462 of the revised manuscript to address this comment. It now reads: “Knowledge and wellbeing are seen to arise through experiences of Country or Land as a living entity [69, 71, 75]. Individuals are seen to draw nurturance from and carry responsibilities to their social and ecological worlds. A deep and enduring connectedness with the natural world – lands, waters, other living things – is commonly understood as fundamental [36, 76]. This ecological view of the interrelatedness of human existence, knowledge, culture, Country or Land and wellbeing [71, 75] is sometimes described as ‘relational wellbeing’ [72, 73] or ‘relationality’ [74].

Reviewer comment: Section 3.2.5: The interface

You talk about ‘Western’ and give an ‘Eastern’ example with the Buddhist notion of mindfulness.  Indigenous Cherokee writer Randy S. Woodley has written about the concept of harmony with the land and how it is the same as the Judeo-Christian concept of shalom. Perhaps this or another, are  a better example as something ‘Western’.

Author response: We do indeed comment on ‘Western’ knowledge systems at several points. However, the clear focus throughout, and in this section, is NOT on the interface between specifically ‘Western’ and non-Indigenous theories. Rather it is on the interface between non-Indigenous and Indigenous theories of wellbeing. Thus, we do not see mention of Buddhist ideas of mindfulness in this section as in any way inconsistent with our approach.

Reviewer comment: Line 529 about the fragmentation of knowledge: Cajete and many others have written about this, add references.

Author response: Two references have been added to the relevant passage on lines 552-558 of the revised manuscript.

Ref 94 is Beam RD. Fragmentation of knowledge: An obstacle to its full utilization. In The Optimum Utilization of Knowledge: Routledge; 2019. p. 160-74.

Ref 95 is Cajete G. Look to the mountain: An ecology of indigenous education: ERIC; 1994.

Reviewer comment: Overall, it is a good and needed work; I would explain the methods better by giving examples of the dialogical process.  There is not enough examples of work outside Australia and New Zealand.

Author response: Thank you for your positive assessment of the overall value of our work.

Reviewer 2 Report

Thank you for the opportunity to read and review this paper. I enjoyed the discussion and agreed wholeheartedly with the significance of the questions being asked and the conclusions of the paper. The paper appears to argue that its ‘methods’ are a systematic review of literature to address key research questions. It then presents its ‘findings’ to answer each of the research questions. The objectives and discussion of this process was somewhat confusing (more below), and not clearly explained or justified. It also feels like this paper is trying to do an awful lot, and to achieve each of these different aims in a systematic way, in not achievable.  

Regarding the focus on indigenous health. This is an important part of the discussion. However, to some extent the framing of the debate seems to suggest we can make generalisation about indigenous peoples, culture or  knowledge’s.  As the authors later argue (on page 10) indigenous populations are diverse and conceptions,  experiences and practices of well-being differ between cultures within and between countries. The methods appear to gloss over this point, rather than pointing to local, place-based understanding of wellbeing,  as some authors argue? Alongside this as the authors note that although they use Durie (a Māori scholar ), the majority of the literature they have drawn on is Australian based, and particularly Torres Straight Islanders.  This is one example of the ways in which this paper appears to draw quite selectively on literatures. This is not an issue per se but does have limitations if the claims here are that a particular selection and sampling method has been used rather than just claiming this is a discussion article which draws out particular thematic areas.  

Materials and methods

I found the discussion on the materials and methods difficult to follow. (I recognise this subheader is prescribed by the journal, and may not be helpful in developing a clear sense of what this research entailed in review papers). I was unsure if this was claiming to be a systematic literature review.  

Page 3 The authors say their aim is to analyse selected literature on indigenous and non- indigenous theories of well-being, and then to present a typology of well-being theory. To develop the well-being typology a literature review ‘method’ was undertaken. Then a narrative review of selected literature was taken place. As the authors note this was necessarily selective and across multiple fields.

Defining the parameters here appear incredibly difficult as well-being is so widely written about. It is certainly clear from the citations used in the paper that there are bodies of literature the authors have not looked at in their review. This is not a criticism; I recognise this is not possible. However, the authors present their approach as being a systematically sampled broad body of literature to then produce a typology. Later, it is noted 333 items were included in the review with 205 indigenous specific documents and 128 non indigenous. This would suggest this sample is not representative of the broader international wellbeing literature.

If this selection process is key to this methods validity, (i.e. some sort of systemic review)  then it would be useful to have tables showing the process, actual articles used, their fields, etc,  and also the different types of theory each one spoke to ?  This would be more consistent with research claiming to be a systematic review.  

If this was not your intention, then I was unsure why there needed to be a discussion of doing a literature review?  

Note:  the second objective is to critically assess the suitability of each type of theory to form effective policy to promote public well-being.

It’s not clear if you only reviewed just these three articles cited? (Page 3 line 128-30).

The explanations of well being they related to indigenous peoples is somewhat clearer and the inclusion of grey literature certainly makes sense. However, as the authors state the majority focused on Aboriginal and Torres Island of perspectives, would this not have been a clearer focus?

The part of the method related to public health promotion and public policy theory  is least clear in terms of the scope and objectives.

The authors allude to their commitment to indigenous methodologies. E.g.  ‘We used Indigenous dialogic methods of open-ended yarning 182 and deep listening’ and  ‘We applied Durie’s principles of mutual respect, shared benefits, human dignity and discovery [36] to undertake the research as a process of collaborative enquiry and 114 mutual capacity building between Indigenous (TM and MS) and non-Indigenous (MF) authors.’ However the reader doesn’t get a sense of how this is relevant to this lietrtaure review, or why. If relevant it need to be explained.    

In terms of the results, the typology of well-being theory is useful. However, none of this is particularly surprising to anyone who has read the contemporary well-being literature.  I would suggest that this this typology would be more useful as a way to frame a literature review, and then as a basis  to  discuss the limitations of these approaches for public policy? Or, as noted above, much more detail is needed about how this typology was produced, and its scope and potential omissions.  

The discussion and recognition of indigenous wellbeing theories is very important within broader discussions of well-being and how it can be more useful a defined and measured. However, with the limited discussion that can be given within this paper I feel is too brief to be useful. The table page 11 shows a number of important points of intersection. However, many of these points are discussed in detail within different literatures. For example, point 2 ecological well-being. There is a wealth of literature discussing how contact with nature is a determinant of mental spiritual and planetary health and well-being in difference countries, contexts, and demographics.  There is also extensive literature for example on how culture provides knowledge values and practices of well-being both from indigenous theory and non-indigenous theory. See for example research available via What works wellbeing website

https://whatworkswellbeing.org/about-wellbeing/how-to-measure-wellbeing/

On page 2 the article outlines the three important interrelated questions.

Number three is less clear, particularly the extent to which theories of change are implicit. The rationale for why a specific focus on social determinants of mental health and/or indigenous health including stress is not clearly articulated. Is the role of stress not important across all populations?

The discussion section is well written and makes lots of important points. I particularly like the way in which this is related to the ways in which governments continues to draw on narrow constructs ( p12 line535). The point that you come to about the need for more flexible, holistic and open approaches to well-being policy that encompasses individual, social, equity and environmental perspective is central, but also one I feel is starting to be recognised across  different literature is on well-being. Similarly, the importance of place based approaches of well-being including in policy is certainly something that is very prevalent in geography and place literatures, particularly those recognising indigenous perspectives. (Page 13 line 572)

The conclusions make really important points.

Overall, there is much in this paper that is excellent. However, the ways it is structured and framed, feel like it limits the discussion of complex and important issues. Perhaps rather than trying to show the systematic nature of the literature-based research, it could be written as a critical review framed around your three excellent questions;  What types of wellbeing theory are available to policy makers? What theory of change are differing types of wellbeing theory likely to present to policy makers, for purposes of application in wellbeing policies? How and to what extent are the theories of change implicit in different types of wellbeing theory?

That way, I fell it would allow you more scope to draw out the key points, and also to show the specified and nuances in the debates, including indigenous well-being theories in Australia, how they differ, and the potential for integration.

Author Response

Reviewer comment: Thank you for the opportunity to read and review this paper. I enjoyed the discussion and agreed wholeheartedly with the significance of the questions being asked and the conclusions of the paper. The paper appears to argue that its ‘methods’ are a systematic review of literature to address key research questions. It then presents its ‘findings’ to answer each of the research questions. The objectives and discussion of this process was somewhat confusing (more below), and not clearly explained or justified. It also feels like this paper is trying to do an awful lot, and to achieve each of these different aims in a systematic way, in not achievable.  

Author response: We thank the reviewer for their positive assessment of the significance our research questions and conclusions.

Here and in several other comments below, the reviewer asserts that, in our methods, we claim to present a ‘systematic’ review of literature to address our research questions but then fail to describe the methods for, or deliver, such a review. We disagree with this criticism.  At no point in our methods section do we use the words ‘systematic’ or ‘systematic review’. On the contrary, the original manuscript explicitly described our approach as a “narrative review of selected literature”, using “purposive selection strategies” to select literature useful for a specific purpose, namely to identify “types of Indigenous and non-Indigenous wellbeing theory”. As Snyder [1] makes clear, literature reviews can take several forms, suited to different purposes and different forms of literature. One such purpose, according to Snyder, is to ‘to evaluate theory or evidence in a certain area or to examine the validity or accuracy of a certain theory or competing theories.” The focus of our review on types of wellbeing theory is consistent with this description. We make clear that, from the outset, the aim of our review was not to review ‘literature on wellbeing’ (an impossible task) but to identify and describe types of wellbeing theory and thus formulate a working typology. In the paragraph starting, “Our selection strategy…” and in the following two paragraphs we set out the steps that we took to purposively select a limited body of literature suited to this task.

We recognise that most literature reviews set out to examine the current state of empirical research and evidence on a particular topic. We acknowledge that what we are doing is somewhat different to this ‘standard’ approach. However, we agree with Snyder that it is equally valid to conduct a literature review to examine the current state of theory of a particular topic. Therefore, we think that our findings as described, and the way we use them to answer our research questions, are legitimate.

Nevertheless, we acknowledge the reviewer’s broader point that the clarity of description and justification of our approach needed to be improved. In addition to the changes made to our methods section in response to reviewer 1’s comments, we made a further change to address these concerns about a lack of clarity from reviewer 2, on p. 3, lines 121-124.  The revised text now reads, “It became obvious that any form of systematic review of this full range of material was not feasible and, indeed, would not serve our purpose. Instead, we decided to undertake a narrative review of a far more limited body of purposively selected literature, with selection strategies designed to pick materials most suited to identifying types of Indigenous and non-Indigenous wellbeing theory.”

[1] Snyder, Hannah. "Literature review as a research methodology: An overview and guidelines." Journal of Business Research, 2019;104: 333-339.

Reviewer comment: Regarding the focus on indigenous health. This is an important part of the discussion. However, to some extent the framing of the debate seems to suggest we can make generalisation about indigenous peoples, culture or knowledges.  As the authors later argue (on page 10) indigenous populations are diverse and conceptions, experiences and practices of well-being differ between cultures within and between countries. The methods appear to gloss over this point, rather than pointing to local, place-based understanding of wellbeing, as some authors argue? Alongside this as the authors note that although they use Durie (a Māori scholar), the majority of the literature they have drawn on is Australian based, and particularly Torres Straight Islanders.  This is one example of the ways in which this paper appears to draw quite selectively on literatures. This is not an issue per se but does have limitations if the claims here are that a particular selection and sampling method has been used rather than just claiming this is a discussion article which draws out particular thematic areas.  

Author response: In section 3.2.4., lines 448-450 the sentence to which the reviewer refers states that, “Whilst conceptions, experiences and practices of wellbeing differ between Indigenous cultures [71] – within or between countries – there are significant shared perspectives [36]. We do not see this as a claim that it is possible or appropriate to make sweeping generalisations about ‘Indigenous peoples, culture or knowledges’. We see it instead as a more nuanced claim, which acknowledges the cultural and epistemological heterogeneity among Indigenous peoples worldwide, but also points to significant commonalities between Indigenous cultures regarding some of the main elements of human wellbeing. This recognition of commonalities among diverse cultures is also reflected in the literature, including articles cited in our revised paper such as Durie (2005), Salmon et al. (2018) and Cajete (2005).

We thank the reviewer for raising a concern that the majority of literature drawn on in the original manuscript was Australian. Therefore, in our revised manuscript and particularly in sections 3.2.4 and 3.2.5 we have taken steps to cite more relevant literature on Indigenous wellbeing from Canada and the United States.

Materials and methods

Reviewer comment: I found the discussion on the materials and methods difficult to follow. (I recognise this subheader is prescribed by the journal, and may not be helpful in developing a clear sense of what this research entailed in review papers). I was unsure if this was claiming to be a systematic literature review.  

Author response: Responses above and changes made to the methods section to improve clarity address this comment.

Reviewer comment: Page 3 The authors say their aim is to analyse selected literature on indigenous and non- indigenous theories of well-being, and then to present a typology of well-being theory. To develop the well-being typology a literature review ‘method’ was undertaken. Then a narrative review of selected literature was taken place. As the authors note this was necessarily selective and across multiple fields.

Defining the parameters here appear incredibly difficult as well-being is so widely written about. It is certainly clear from the citations used in the paper that there are bodies of literature the authors have not looked at in their review. This is not a criticism; I recognise this is not possible. However, the authors present their approach as being a systematically sampled broad body of literature to then produce a typology. Later, it is noted 333 items were included in the review with 205 indigenous specific documents and 128 non indigenous. This would suggest this sample is not representative of the broader international wellbeing literature.

Author response: As noted in responses above, we do not present our approach as ‘a systematically sampled broad body of literature’ and our purposive methods of selecting literature did not set out to create a sample ‘representative of the broader international wellbeing literature’. Therefore, we think the implied assertion that the work is lacking in rigour because it does not consider some elements of this broader literature is misplaced.  However, once again, the reviewer’s comments indicate a need to clarify the nature and purposes of our review process. The changes we have made in the methods section, pp. 3-5, as noted above address this concern. The reason that a larger number of Indigenous-specific documents were reviewed is because the selection process for that part of the review included both academic and grey literature (which the reviewer describes below as making sense) and spanned across four jurisdictions.

Reviewer comment: If this selection process is key to this methods validity, (i.e. some sort of systemic review)  then it would be useful to have tables showing the process, actual articles used, their fields, etc,  and also the different types of theory each one spoke to ?  This would be more consistent with research claiming to be a systematic review. If this was not your intention, then I was unsure why there needed to be a discussion of doing a literature review?  

Author response: We believe that our responses to earlier comments above address this concern.

Reviewer comment: Note:  the second objective is to critically assess the suitability of each type of theory to form effective policy to promote public well-being.

It’s not clear if you only reviewed just these three articles cited? (Page 3 line 128-30).

Author response: It is not quite clear to us what point the reviewer is making here. If the reviewer is referring to the sentence on line 131-133 where we say’ “we reviewed several recent journal articles explicitly focused on wellbeing theory, where well-known authors in the field summarized theory types [18, 37-39]”, the articles cited are examples of the material examined.

Reviewer comment: The explanations of well being they related to indigenous peoples is somewhat clearer and the inclusion of grey literature certainly makes sense. However, as the authors state the majority focused on Aboriginal and Torres Island of perspectives, would this not have been a clearer focus?

Author response: We have acknowledged that literature on Indigenous views of wellbeing within the original manuscript was too heavily weighted toward Australian literature and have taken steps to correct this, as noted in our response to earlier comments.

Reviewer comment: The part of the method related to public health promotion and public policy theory  is least clear in terms of the scope and objectives.

Author response: Thank you.

Reviewer comment: The authors allude to their commitment to indigenous methodologies. E.g.  ‘We used Indigenous dialogic methods of open-ended yarning 182 and deep listening’ and  ‘We applied Durie’s principles of mutual respect, shared benefits, human dignity and discovery [36] to undertake the research as a process of collaborative enquiry and 114 mutual capacity building between Indigenous (TM and MS) and non-Indigenous (MF) authors.’ However the reader doesn’t get a sense of how this is relevant to this literature review, or why. If relevant it needs to be explained.    

Author response: We have made changes to the methods section lines 174-186, as noted in response to comments from reviewer 1, which clarify what our dialogic methods of yarning at the interface’ entailed and how we applied them to analyse literature to firstly discern types of wellbeing theory and then to assess the suitability of each type to inform wellbeing policy.

Reviewer comment: In terms of the results, the typology of well-being theory is useful. However, none of this is particularly surprising to anyone who has read the contemporary well-being literature.  I would suggest that this this typology would be more useful as a way to frame a literature review, and then as a basis  to  discuss the limitations of these approaches for public policy? Or, as noted above, much more detail is needed about how this typology was produced, and its scope and potential omissions.  

Author response: We agree that there is much literature that, in one way or another, discusses and perhaps seeks to define some different types of wellbeing theory. However, in our view there is little literature available that sets out a more formal typology of wellbeing theories incorporating biomedical and behavioural conceptions and combining non-Indigenous and Indigenous theory. We have added more detail in the methods section about how the typology was produced.  

Reviewer comment: The discussion and recognition of indigenous wellbeing theories is very important within broader discussions of well-being and how it can be more useful a defined and measured. However, with the limited discussion that can be given within this paper I feel is too brief to be useful. The table page 11 shows a number of important points of intersection. However, many of these points are discussed in detail within different literatures. For example, point 2 ecological well-being. There is a wealth of literature discussing how contact with nature is a determinant of mental spiritual and planetary health and well-being in difference countries, contexts, and demographics.  There is also extensive literature for example on how culture provides knowledge values and practices of well-being both from indigenous theory and non-indigenous theory. See for example research available via What works wellbeing website

https://whatworkswellbeing.org/about-wellbeing/how-to-measure-wellbeing/

Author response: The reviewer makes a valid point, that some of the points made in Table 2 are indeed discussed in more detail in different bodies of literature such as that examining contact with nature as a determinant of health. However, the purpose of our article is to look across a number of theoretic conceptions of wellbeing, rather than to delve deeply into one or two conceptions. (There is no doubt that each of the theory types noted in Table 1 would ‘lead off’ into an extended body of related literature.) Thus, in keeping with this overall purpose, the aim of Table 2 is to summarise a number of points of intersection and, perhaps, possible theoretic integration, between Indigenous and non-Indigenous conceptions of wellbeing, rather than to look deeply into one or two. In introducing Table 2, on lines 517-519, we state explicitly that “Here we will not attempt to further tease out the implications of these points of intersection, but simply present them as opportunities for further research and dialogue”.

Reviewer comment: On page 2 the article outlines the three important interrelated questions.

Number three is less clear, particularly the extent to which theories of change are implicit. The rationale for why a specific focus on social determinants of mental health and/or indigenous health including stress is not clearly articulated. Is the role of stress not important across all populations?

Author response: Thanks for this comment. We took the view that theories of wellbeing as such, being primarily concerned with questions about what wellbeing is, are unlikely to be concerned with articulating an explicit theory of change as found in public policies; namely one that makes a “‘predictive assumption about the relationship between desired changes and the [policy] actions that may produce those changes’ [31, p. 1]”. Thus, the way a theory of wellbeing would inform a theory of change for policy is likely to remain implicit, until someone sets out to interpret the theory in this way. That is why we frame question 3 as we have. We set out our interpretations on this point in our results section. The rationale for a focus on social determinants of health and stress as a mediator of social-environmental impacts on mental health is simply the very large body of evidence indicating the importance of social factors in determining population mental health and wellbeing outcomes. Thus, the premise from the beginning is that, if we wish to pursue policy to promote mental health and wellbeing, then, to be effective, such policy must take account of and address social determinants of health and give consideration to the role of stress as a mediator of these effects. The role of stress is indeed important across all populations and we believe the wording of Q3 reflects that view.

Reviewer comment: The discussion section is well written and makes lots of important points. I particularly like the way in which this is related to the ways in which governments continues to draw on narrow constructs ( p12 line535). The point that you come to about the need for more flexible, holistic and open approaches to well-being policy that encompasses individual, social, equity and environmental perspective is central, but also one I feel is starting to be recognised across  different literature is on well-being. Similarly, the importance of place based approaches of well-being including in policy is certainly something that is very prevalent in geography and place literatures, particularly those recognising indigenous perspectives. (Page 13 line 572)

Author response: Thank you for this positive assessment of our discussion. Again, the reviewer has a point about broader bodies of literature on holistic approaches to wellbeing policy and place-based approaches. Again, our response is that, in this kind of article, spanning across a range of theory and policy perspectives on wellbeing, it is not possible to reflect the depth of literature linked to each one of these perspectives. 

The conclusions make really important points.

Author response: Thank you.

Reviewer comment: Overall, there is much in this paper that is excellent. However, the ways it is structured and framed, feel like it limits the discussion of complex and important issues. Perhaps rather than trying to show the systematic nature of the literature-based research, it could be written as a critical review framed around your three excellent questions;  What types of wellbeing theory are available to policy makers? What theory of change are differing types of wellbeing theory likely to present to policy makers, for purposes of application in wellbeing policies? How and to what extent are the theories of change implicit in different types of wellbeing theory?

That way, I fell it would allow you more scope to draw out the key points, and also to show the specified and nuances in the debates, including indigenous well-being theories in Australia, how they differ, and the potential for integration.

Author response: Thanks for the suggestion about a possible more substantial change in the nature of the proposed article. However, we believe our approach is justified and worthwhile, and we hope that changes made have addressed the reviewer’s concerns.